# Provable Guarantees for Generative Behavior Cloning: Bridging Low-Level Stability and High-Level Behavior

**Adam Block**
MIT
Cambridge, MA
ablock@mit.edu

**Ali Jadbabaie**
MIT
Cambridge, MA
jadbabai@mit.edu

**Daniel Pfrommer**
MIT
Cambridge, MA
dpfrom@mit.edu

**Max Simchowitz**
MIT
Cambridge, MA
msimchow@csail.mit.edu

**Russ Tedrake**
MIT
Cambridge, MA
russt@mit.edu

## Abstract

We propose a theoretical framework for studying behavior cloning of complex expert demonstrations using generative modeling. Our framework invokes low-level controllers - either learned or implicit in position-command control - to stabilize imitation around expert demonstrations. We show that with (a) a suitable low-level stability guarantee and (b) a powerful enough generative model as our imitation learner, pure supervised behavior cloning can generate trajectories matching the per-time step distribution of essentially arbitrary expert trajectories in an optimal transport cost. Our analysis relies on a stochastic continuity property of the learned policy we call "total variation continuity" (TVC). We then show that TVC can be ensured with minimal degradation of accuracy by combining a popular data-augmentation regimen with a novel algorithmic trick: adding augmentation noise at execution time. We instantiate our guarantees for policies parameterized by diffusion models and prove that if the learner accurately estimates the score of the (noise-augmented) expert policy, then the distribution of imitator trajectories is close to the demonstrator distribution in a natural optimal transport distance. Our analysis constructs intricate couplings between noise-augmented trajectories, a technique that may be of independent interest. We conclude by empirically validating our algorithmic recommendations, and discussing implications for future research directions for better behavior cloning with generative modeling.

## 1 Introduction

Training dynamic agents from datasets of expert examples, known as *imitation learning*, promises to take advantage of the plentiful demonstrations available in the modern data environment, in an analogous manner to the recent successes of language models conducting unsupervised learning on enormous corpora of text [67, 70]. Imitation learning is especially exciting in robotics, where mass stores of pre-recorded demonstrations on Youtube [1] or cheaply collected simulated trajectories [42, 20] can be converted into learned robotic policies.

For imitation learning to be a viable path toward generalist robotic behavior, it needs to be able to both represent and *execute* the complex behaviors exhibited in the demonstrated data. An approach that has shown tremendous promise is *generative behavior cloning:* fitting generative models, such

37th Conference on Neural Information Processing Systems (NeurIPS 2023).

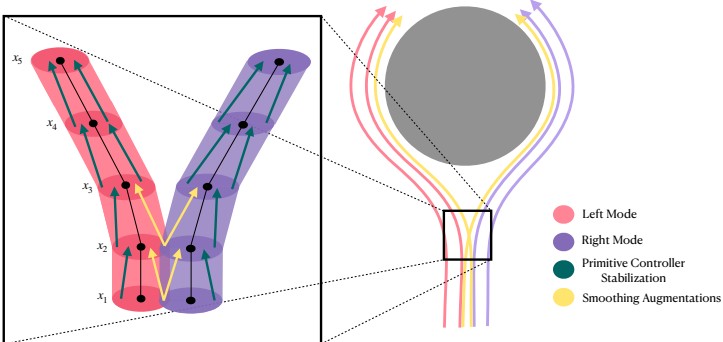

Figure 1: Consider demonstration trajectories exhibiting two modes: a "go left" and "go right" mode around an obstacle depicted in red and purple, respectively. To avoid compounding error, we imitate sequences of simple low-level feedback controllers we call "primitive controllers", not simply raw actions. Intuitively, primitive controllers provide "tubes" around each demonstration trajectory where the system can be stabilized. Depicted in yellow, our data-noising procedure described below "fills in the gaps" in the demonstration, switching between modes in a well-behaved manner, and whilst allowing the primitive controllers to manage the stabilization.

as diffusion models [2, 19, 33], to expert demonstrations with pure supervised learning. In this paper, we ask: **When can generative behavior cloning imitate arbitrarily complex expert behavior?**

In this paper, we are interested in how *algorithmic choices* interface with the *dynamics of the agent's environment* to render imitation possible. The key challenge separating imitation learning from vanilla supervised learning is one of *compounding error*: when the learner executes the trained behavior in its environment, small mistakes can accumulate into larger ones; this in turn may bring the agent to regions of state space not seen during training, leading to larger-still deviations from intended trajectories. Without the strong requirement that the learner can interactively query the expert at new states [40, 57], it is well understood that ensuring some form of *stability* in the imitation learning procedure is indispensable [68, 27, 50]. While many natural notions of stability exist for simple behaviors, how to *enforce* stability when imitating more complex behaviors remains an open question. Multi-modal trajectories present a key example of this challenge: consider a robot navigating around an obstacle; because there is no difference between navigating around the object to the right and around to the left, the dataset of expert trajectories may include examples of both options. This bifurcation of good trajectories can make it difficult for the agent to effectively choose which direction to go, possibly even causing the robot to oscillate between directions and run into the object. [19]. Moreover, human demonstrators correlate current actions with the past in order to *commit* to either a right or left path, which makes even formulating the idea of an "expert *policy*" a conceptually challenging one. Lastly, bifurcations are necessarily *incompatible* with previous notions of stability derived from classical control theory [68, 27, 50]. **In this work, we investigate how these strong and often unrealistic assumptions on the expert policy can be replaced by practical (and often realistic) assumptions on available algorithms.**

## 1.1 Contributions.

As in previous work, we formalize behavior cloning in two stages: at *train-time*, we learn a map from observations to distributions over actions, supervised by (state, action)-pairs from expert demonstrations coming from $N_{\exp}$ independent expert trajectories, while at *test-time*, the learned map, or *policy*, is executed on random initial states (distributed identically to initial training states). Following the schematic of existing theoretical analyses of behavior cloning [68, 50, 27], we demonstrate that a policy trained by minimizing a certain supervised learning objective on expert demonstrations induces trajectories that approximate those of expert demonstrations. Our work considers a significantly more general setting than past theoretical literature, and one which reflects the strength of *generative models* for imitation. One corollary of our key contributions is summarized in the following informal statement. The main technical insights leading to the proof of the theorem are detailed in the bullet points below it, and depicted in Figure 1.

**Theorem (informal).** *Consider a generative behavior cloner $\hat{\pi}$ that learns to predict sequences of expert actions on horizon $H$, **along with low-level controllers that locally stabilize the trajectories**. Then, with a suitable data noising strategy, for all times $h \leq H$,*

$$\mathbb{P}[\text{expert \& imitator trajectories disagree at some time } h \text{ by } \geq \varepsilon]$$
$$\leq \mathcal{O}_{\text{Iss}}\left(H\varepsilon + \tfrac{1}{\varepsilon^2}\sum_h \mathbb{E}_{\text{expert},h}\left[\mathcal{W}_1(\mathbb{P}_{\text{expert actions}}, \mathbb{P}_{\text{imitator actions}})\right]\right)$$

*where $\mathbb{E}_{\text{expert},h}[\mathcal{W}_1(\mathbb{P}_{\text{expert actions}}, \mathbb{P}_{\text{imitator actions}})]$ denotes a 1-Wasserstein distance in an appropriate metric between the conditional distribution over expert and imitator actions given the observation at time step $h$, and where $\mathcal{O}_{\text{Iss}}$ hides constants depending polynomially on the stability properties of the low-level controllers, defined formally in Section 3.1.*

We now detail the key ingredients of our results.

1. We imitate stochastic demonstrators that may exhibit both complex correlations between actions in their trajectories (e.g. be non-Markovian) and multi-modal behavior. The natural object to imitate in this setting is the conditional probability distribution of expert actions given recent states, but marginalized over past states. We require said conditional action distribution to be learnable by a **generative model**, but otherwise arbitrarily complex: in particular, the conditional distribution of an expert actions given the state can be discontinuous (in any natural distance metric) as a function of state, as in the bifurcation depicted in Figure 1*(right)*.

2. We obtain **rigorous, theoretical guarantees** and without requiring either interactive data collection (e.g. DAGGER [57, 40]), or access to gradients of the expert policy (as in TASIL[50]). Instead, we replace these assumption with an oracle, described below, which **synthesizes stabilizing, low-level policies** along training demonstrations—the green arrows in Figure 1*(left)*. This mirrors recent work on generative behavior cloning that find that providing state-commands through inverse dynamics controllers [33, 2] or position-command controllers of end effectors [19] leads to substantially improved performance.

3. We also apply a subtle-yet-significant modification to a popular **data noising** strategy, which we show yields both theoretical and empirical beneifits. Data noising ensures a helpful property we denote *total variation continuity* that interpolates between modes in probability space (without naively averaging their trajectories in world space). This effectively "fills in the missing gaps" in bifurcations, as indicated by yellow arrows in Figure 1.

Our main results, Theorems 1 and 2, are reductions from imitation of complex expert trajectories to supervised generative learning of a specific conditional distribution. For concreteness, Theorem 3 instantiates the generative modeling with Denoising Diffusion Probabilistic Models (DDPMs) of sufficient regularity and expressivity (as investigated empirically in [19, 48, 26]), and establishes end-to-end guarantees for imitation of complex trajectories with sample complexity polynomial in relevant problem parameters. Our analysis framework exposes that any sufficient powerful generative learner obtains similar guarantees. Finally, we empirically validate the benefits of our proposed smoothing strategy in simulated robotic manipulation tasks. We now summarize the algorithmic choices and analytic ideas that facilitate our reduction.

**Abridged Related Work.** Due to space, we defer a full comparison to past work to Appendix B. DDPMs, proposed in [29, 60], along with their relatives have seen success in image generation [62, 55], along with imitation learning (without data augmentation) [33, 19, 48], which is the starting point of our work. Smoothing data augmentation is ubiquitous in modern imitation learning [40] and our approach corresponds to that of [36] but with noise added at inference time. Despite the benefits of adaptive data collection [58, 40], adaptive demonstrations are more expensive to collect. Previous analyses of imitation learning without adaptive data collection have focused on classical control-theoretic notions of stability, notably incremental stability, [68, 27, 50], which require continuity, Markovianity, and often determinism, and preclude the bifurcations permitted in our setting.

## 2 Setting

**Notation and Preliminaries.** Appendix A gives a full review of notation. Bold lower-case (resp. upper-case) denote vectors (resp. matrices). We abbreviate the concatenation of sequences via $\mathbf{z}_{1:n} = (\mathbf{z}_1, \ldots, \mathbf{z}_n)$. Norms $\|\cdot\|$ are Euclidean for vectors and operator norms for matrices unless

otherwise noted. Rigorous probability-theoretic preliminaries are provided in Appendix F. In short, all random variables take values in Polish spaces $\mathcal{X}$ (which include real vector spaces), the space of Borel distributions on $\mathcal{X}$ is denoted $\Delta(\mathcal{X})$. We rely heavily on ***couplings*** from optimal transport theory: given measures $X \sim \mathsf{P}$ and $X' \sim \mathsf{P}'$ on $\mathcal{X}$ and $\mathcal{X}'$ respectively, $\mathscr{C}(\mathsf{P},\mathsf{P}')$ denotes the space of joint distributions $\mu \in \Delta(\mathcal{X} \times \mathcal{X}')$ called "couplings" such that $(X, X') \sim \mu$ has marginals $X \sim \mathsf{P}$ and $X' \sim \mathsf{P}$. $\Delta(\mathcal{X} \mid \mathcal{Y})$ denotes the space of conditional probability distributions $\mathsf{Q} : \mathcal{Y} \to \Delta(\mathcal{X})$, formally called probability ***kernels*** ; Appendix F rigorously justifies that, in our setting, all conditional distributions can be expressed as kernels (which we do throughout the paper without comment). Finally $\mathbf{I}_\infty(\mathcal{E})$ denotes the indicator taking value 1 if $\mathcal{E}$ is true and $\infty$ otherwise.

**Dynamics and Demonstrations.** We consider a discrete-time, control system with states $\mathbf{x}_t \in \mathcal{X} := \mathbb{R}^{d_x}$, and inputs $\mathbf{u}_t \in \mathcal{U} := \mathbb{R}^{d_u}$, obeying the following nonlinear dynamics

$$\mathbf{x}_{t+1} = f(\mathbf{x}_t, \mathbf{u}_t), \quad t \geq 1. \tag{2.1}$$

Given length $T \in \mathbb{N}$, we call sequences $\boldsymbol{\rho}_T = (\mathbf{x}_{1:T+1}, \mathbf{u}_{1:T}) \in \mathscr{P}_T := \mathcal{X}^{T+1} \times \mathcal{U}^T$ ***trajectories***. For simplicity, we assume that (2.1) is deterministic and address stochastic dynamics in Appendix N. Though the dynamics are Markov and deterministic, we consider a stochastic and possibly *non-Markovian* demonstrator, which allows for the multi-modality described in the Section 1.

**Definition 2.1** (Expert Distribution). Let $\mathcal{D}_{\exp} \in \Delta(\mathscr{P}_T)$ denote an ***expert distribution*** over trajectories to be imitated. $\mathcal{D}_{\mathbf{x}_1}$ denotes the distribution of $\mathbf{x}_1$ under $\boldsymbol{\rho}_T = (\mathbf{x}_{1:T+1}, \mathbf{u}_{1:T}) \sim \mathcal{D}_{\exp}$.

**Primitive Controllers.** Our approach is to imitate not just actions, but simple local *control policies*. In the body of this paper, we consider affine mappings $\mathcal{X} \to \mathcal{U}$ (redundantly) parameterized as $\mathbf{x} \mapsto \bar{\mathbf{u}} + \bar{\mathbf{K}}(\mathbf{x} - \bar{\mathbf{x}})$; we call these ***primitive controllers***, denoted with $\kappa = (\bar{\mathbf{u}}, \bar{\mathbf{x}}, \bar{\mathbf{K}}) \in \mathcal{K}$. We describe the synthesis of these controllers in Appendix D , and extend our results to general families of parameterized controllers in Appendix E. We argue in Appendix E that primitive controllers are in fact standard practice, and implicit via robotic position control in many applications of diffusion to robotic behavior cloning.

**Chunking Policies and Indices.** The expert distribution $\mathcal{D}_{\exp}$ may involve non-Markovian sequences of actions. We imititate these sequences via ***chunking policies***. Fix a ***chunk length*** $\tau_{\mathrm{chunk}} \in \mathbb{N}$ and ***observation length*** $\tau_{\mathrm{obs}} \leq \tau_{\mathrm{chunk}}$, and define time indices $t_h = (h-1)\tau_{\mathrm{chunk}} + 1$. For simplicity, we assume $\tau_{\mathrm{chunk}}$ divides $T$, and set $H = T/\tau_{\mathrm{chunk}}$. Given a $\boldsymbol{\rho}_T \in \mathscr{P}_T$, define the ***trajectory-chunks*** and ***observation*** chunks

$$\mathsf{s}_h := (\mathbf{x}_{t_{h-1}:t_h}, \mathbf{u}_{t_{h-1}:t_h-1}) \in \mathcal{S} := \mathscr{P}_{\tau_{\mathrm{chunk}}} \qquad \text{(trajectory-chunks)}$$

$$\mathsf{o}_h := (\mathbf{x}_{t_h-\tau_{\mathrm{obs}}+1:t_h}, \mathbf{u}_{t_h-\tau_{\mathrm{obs}}+1:t_h-1}) \in \mathcal{O} := \mathscr{P}_{\tau_{\mathrm{obs}}-1} \qquad \text{(observation-chunks)}$$

for $h > 1$, and $\mathsf{s}_1 = \mathsf{o}_1 = \mathbf{x}_1$ (for simplicity, we embed $\mathsf{o}_1$ into $\mathscr{P}_{\tau_{\mathrm{obs}}-1}$ via zero-padding). We call $\tau_{\mathrm{chunk}}$-length sequences of primitive controllers ***composite actions***

$$\mathsf{a}_h = \kappa_{t_h:t_{h-1}} \in \mathcal{A} := \mathcal{K}^{\tau_{\mathrm{chunk}}}. \qquad \text{(composite actions)}$$

A ***chunking policy*** $\pi = (\pi_h)$ consists of functions $\pi_h$ mapping observation-chunks $\mathsf{o}_h$ to distributions $\Delta(\mathcal{A})$ over composite actions and interacting with the dynamics (2.1) by $\mathsf{a}_h = \kappa_{t_h:t_{h-1}} \sim \pi_h(\mathsf{o}_h)$, and executing $\mathbf{u}_t = \kappa_t(\mathbf{x}_t)$. We let $d_{\mathcal{A}} = \tau_{\mathrm{chunk}}(d_x + d_u + d_x d_u)$ denote the dimension of the space $\mathcal{A}$ of composite actions. The chunking scheme is represented in Figure 2, demonstrating the rationale for using primitive controllers over open-loop actions. Remark C.1 describes our rationale for studying *states* over generic observations, and considering time-dependent policies.

**Desideratum.** The quality of imitation of a deterministic policy is naturally measured in terms of step-wise closeness of state and action [68, 50]. With stochastic policies, however, two rollouts of even the same policy can visit different states. We propose measuring *distributional closeness* via *couplings* introduced in the preliminaries above. We define the following losses, focusing on the *marginal distributions* between trajectories.

**Definition 2.2.** Given $\varepsilon > 0$ and a (chunking) policy $\pi$, the (marginal distribution) imitation loss is

$\mathcal{L}_{\mathrm{marg},\varepsilon}(\pi) := \max_{t \in [T]} \inf_\mu \mathbb{P}_\mu \left[ \max \left\{ \|\mathbf{x}_{t+1}^{\exp} - \mathbf{x}_{t+1}^\pi\|, \|\mathbf{u}_t^{\exp} - \mathbf{u}_t^\pi\| \right\} > \varepsilon \right]$, where the infimum is over all couplings $\mu$ between the distribution of $\boldsymbol{\rho}_T$ under $\mathcal{D}_{\exp}$ and that induced by the policy $\pi$ as described above, such that $\mathbb{P}_\mu[\mathbf{x}_1^{\exp} = \mathbf{x}_1^\pi] = 1$.

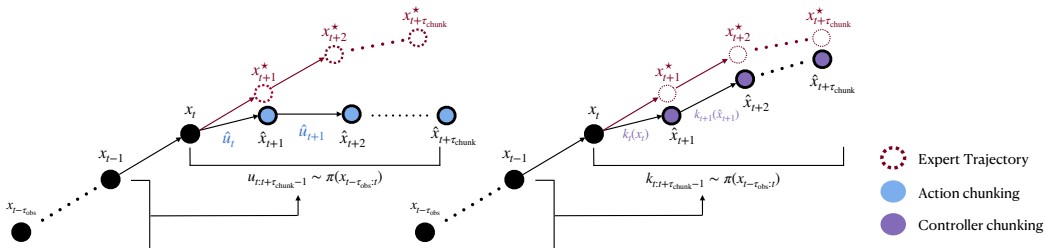

Figure 2: Graphical comparison of an action-chunk based policy (left) as described in [19], versus the primitive-controller chunking policy (right) proposed in this paper. The primitive controller paradigm allows for stabilizing back to the original expert trajectory, whereas using generated actions in an open-loop fashion may cause divergence from the expert in the presence of unstable system dynamics. We refer to **composite actions** as the sequence of primitve controllers given on the right.

Under stronger conditions (whose necessity we establish), we can also imitate joint distributions over actions (Appendix J). Observe that $\mathcal{L}_{\mathrm{fin},\varepsilon} \leq \mathcal{L}_{\mathrm{marg},\varepsilon}$, and that both losses are equivalent to Wasserstein-type metrics on bounded domains. These losses are also equivalent to Lévy-Prokhorov metrics [64] under re-scaling of the Euclidean metric (even for unbounded domains), and also correspond to total variation analogues of shifted Renyi divergences [5, 6]. While empirically evaluating these infima over couplings is challenging, $\mathcal{L}_{\mathrm{marg},\varepsilon}$ upper bounds the difference in expectation between any bounded and Lipschitz control cost decomposing across time steps, states and inputs, and $\mathcal{L}_{\mathrm{fin},\varepsilon}$ upper bounds differences in final-state costs; see Appendix J for discussion.

**Diffusion Models.** Our analysis provides imitiation guarantees when chunking policies $\pi_h$ select $\mathsf{a}_h$ via a sufficiently accurate generative model. Given their recent success, instantiate our analysis for the popular Denoising Diffusion Probabilistic Models (DDPM) framework [18, 41] that allows the learner to sample from a density $q \in \Delta(\mathbb{R}^d)$ assuming that the *score* $\nabla \log q$ is known to the learner. More precisely, suppose the learner is given an observation $\mathsf{o}_h$ and wishes to sample $\mathsf{a}_h \sim q(\cdot|\mathsf{o}_h)$ for some family of probability kernels $q(\cdot|\cdot)$. A DDPM starts with some $\mathsf{a}_h^0$ sampled from a standard Gaussian noise and iteratively "denoises" for each DDPM-time step $0 \leq j < J$:

$$\mathsf{a}_h^j = \mathsf{a}_h^{j-1} - \alpha \cdot \mathsf{s}_{\theta,h}(\mathsf{a}_h^{j-1}, \mathsf{o}_h, j) + 2 \cdot \mathcal{N}(0, \alpha^2 \mathbf{I}), \tag{2.2}$$

where $\mathsf{s}_{\theta,h}(\mathsf{a}_h^j, \mathsf{o}_h, j)$ estimates the true score $\mathsf{s}_{\star,h}(\mathsf{a}_h, \mathsf{o}_h, \alpha j)$, formally defined for any continuous argument $t \leq J\alpha$ to be $\mathsf{s}_{\star,h}(\mathsf{a}, \mathsf{o}_h, t) := \nabla_{\mathsf{a}} \log q_{h,[t]}^{\star}(\mathsf{a} \mid \mathsf{o}_h)$, where $q_{h,[t]}^{\star}(\cdot|\mathsf{o}_h)$ is the distribution of $e^{-t}\mathsf{a}_h^{(0)} + \sqrt{1 - e^{-2t}}\gamma$ with $\mathsf{a}_h^{(0)}$ is sampled from the target distribution we which to sample from, and $\gamma \sim \mathcal{N}(0, \mathbf{I})$ is a standard Gaussian. We denote by $\mathtt{DDPM}(\mathsf{s}_\theta, \mathsf{o}_h)$ the law of $\mathsf{a}_h^J$ sampled according to the DDPM using $\mathsf{s}_\theta(\cdot, \mathsf{o}_h, \cdot)$ as a score estimator. Preliminaries on DPPMs are detailed in Appendix L.

## 3 Conditional sampling with stabilization suffices for behavior cloning

We show that trajectories of the form given in Definition 2.1 can be efficiently imitated if (a) we are given a *synthesis oracle*, described below, that produces low-level control policies that locally stabilize chunks of the trajectory with primitive controllers and (b) we can learn to generate certain appropriate distributions over composite actions, i.e. sequences of primitive controllers. All the following results apply to affine primitive controllers introduced in Section 2 and assume that the system dynamics are second-order smooth and locally stabilizable. In Appendix E, we show that our results still hold with general families of parametric primitive controllers, provided that these controllers induce the same local stability guarantee.

**The synthesis oracle.** We say primitive controller (cf. Section 2) $\kappa_{1:T} \in \mathcal{K}^T$ is *consistent with* a trajectory $\boldsymbol{\rho} = (\mathbf{x}_{1:T+1}, \mathbf{u}_{1:T}) \in \mathscr{P}_T$ if $\bar{\mathbf{x}}_t = \mathbf{x}_t$ and $\bar{\mathbf{u}}_t = \mathbf{u}_t$ for all $t \in [T]$; note that this implies that $\kappa_t(\mathbf{x}_t) = \mathbf{u}_t$ for all $t$. A *synthesis oracle* $\mathtt{synth}$ maps $\mathscr{P}_T \to \mathcal{K}^T$ such that, for all $\boldsymbol{\rho}_T \in \mathscr{P}_T$, $\kappa_{1:T} = \mathtt{synth}(\boldsymbol{\rho}_T)$ is consistent with $\boldsymbol{\rho}_T$. For our theory, we assume access to a synthesis oracle at training time, and assume the ability to estimate conditional distributions over joint sequences of primitive controllers; Appendix K explains how this can be implemented by solving Ricatti equations if dynamics are known (e.g. in a simulator), smooth, and stabilizable. In our experimental environment, control inputs are desired robot configurations, which the simulated

robot executes by applying feedback gains. As discussed in Appendix E, learned or hand-coded low-level controllers are popular in practical implementations of generative behavior cloning. We discuss the merits of studying imitation learning with a synthesis oracle in depth in Appendix C.3.

**Notions of distance.** While restricting ourselves to affine primitive controllers, our approximation error of generative behavior cloner is measured in terms of optimal transport distances that use the following "maximum distance." Given two composite actions $\mathsf{a} = (\bar{\mathbf{u}}_{1:\tau_{\mathrm{chunk}}}, \bar{\mathbf{x}}_{1:\tau_{\mathrm{chunk}}}, \bar{\mathbf{K}}_{1:\tau_{\mathrm{chunk}}})$ and $\mathsf{a}' = (\bar{\mathbf{u}}'_{1:\tau_{\mathrm{chunk}}}, \bar{\mathbf{x}}'_{1:\tau_{\mathrm{chunk}}}, \bar{\mathbf{K}}'_{1:\tau_{\mathrm{chunk}}})$, we define

$$\mathsf{d}_{\max}(\mathsf{a}, \mathsf{a}') := \max_{1 \leq k \leq \tau_{\mathrm{chunk}}} (\|\bar{\mathbf{u}}_k - \bar{\mathbf{u}}'_k\| + \|\bar{\mathbf{x}}_k - \bar{\mathbf{x}}'_k\| + \|\bar{\mathbf{K}}_k - \bar{\mathbf{K}}'_k\|). \tag{3.1}$$

Distances between policies are defined via natural optimal transport costs. Given two policies $\pi = (\pi_h), \pi' = (\pi'_h)$ and observation chunk $\mathsf{o}_h$, we define an induced optimal transport cost

$$\Delta_\varepsilon(\pi_h(\mathsf{o}_h), \pi'_h(\mathsf{o}_h)) := \inf_\mu \mathbb{P}_{(\mathsf{a}_h, \mathsf{a}'_h) \sim \mu} [\mathsf{d}_{\max}(\mathsf{a}_h, \mathsf{a}'_h) > \varepsilon],$$

where the $\inf_\mu$ denotes the infinum over all couplings between $\mathsf{a}_h \sim \pi_h(\mathsf{o}_h)$ and $\mathsf{a}'_h \sim \pi'_h(\mathsf{o}_h)$. $\Delta_\varepsilon$ corresponds to a relaxed Lévy-Prokhorov metric [64], and can always be bounded, via Markov's inequality, by $\Delta_\varepsilon(\pi_h, \pi'_h \mid \mathsf{o}_h) \leq \frac{1}{\varepsilon} \mathcal{W}_{1,\mathsf{d}_{\max}}(\pi_h(\mathsf{o}_h), \pi'_h(\mathsf{o}_h))$, where $\mathcal{W}_{1,\mathsf{d}_{\max}}(\pi_h(\mathsf{o}_h), \pi'_h(\mathsf{o}_h))$ denotes the 1-Wasserstein distance between $\mathsf{a}_h \sim \pi_h(\mathsf{o}_h)$ and $\mathsf{a}'_h \sim \pi'_h(\mathsf{o}_h)$.

### 3.1 Incremental Stability and the Synthesis Oracle.

We assume that synthesis oracle above produces *incrementally stabilizing* control gains, in the sense first proposed by [9]. Incremental stability has emerged as a natural desirable property for imitation limitation [50, 68, 27], because it forces the expert to be robust to small perturbations of their policy. We now supply a formal definition. Given a primitive controller $\kappa : \mathbb{R}^{d_x} \to \mathbb{R}^{d_u}$, define the closed loop dynamic map $f_{\mathrm{cl},\kappa}(\mathbf{x}, \delta\mathbf{u}) := f(\mathbf{x}, \kappa(\mathbf{x}) + \delta\mathbf{u})$. Thus, composite action $\mathsf{a}$ is *consistent* with a trajectory chunk $\mathsf{s} = (\mathbf{x}_{1:\tau_{\mathrm{chunk}}+1}, \mathbf{u}_{1:\tau})$ if $\mathbf{x}_{t+1} = f_{\mathrm{cl},\kappa_t}(\mathbf{x}, \mathbf{0})$ for $1 \leq t \leq \tau_{\mathrm{chunk}}$.[1]

**Definition 3.1** (Time-Varying Incremental Stability). Let $\gamma(\cdot)$ be a class $\mathsf{K}$ function, $\beta(\cdot, \cdot)$ be class $\mathsf{KL}$ function, and let $\mathsf{a} = (\kappa_1, \kappa_2, \ldots, \kappa_\tau)$ denote a sequence of primitive controllers (i.e. a composite action when $\tau = \tau_{\mathrm{chunk}}$). Given a sequence of input perturbations $\delta\mathbf{u}_{1:\tau} \in (\mathbb{R}^{d_u})^\tau$ and initial condition $\boldsymbol{\xi} \in \mathbb{R}^{d_x}$, let $\mathbf{x}^{\mathsf{a}}_{i+1}(\delta\mathbf{u}_{1:\tau}, \boldsymbol{\xi}) = f_{\mathrm{cl},\kappa_i}(\mathbf{x}^{\mathsf{a}}_i(\delta\mathbf{u}_{1:\tau}, \boldsymbol{\xi}), \delta\mathbf{u}_i)$, with $\mathbf{x}^{\mathsf{a}}_1 = \boldsymbol{\xi}$. We say that composite action $\mathsf{a}$ is time-varying incrementally input-to-state stable (t-Iss) with moduli $\gamma(\cdot), \beta(\cdot, \cdot)$ if for all $\boldsymbol{\xi}, \boldsymbol{\xi}' \in \mathbb{R}^{d_x}, 0 \leq i \leq \tau, \|\mathbf{x}^{\mathsf{a}}_i(\mathbf{0}_{1:\tau}, \boldsymbol{\xi}) - \mathbf{x}^{\mathsf{a}}_i(\delta\mathbf{u}_{1:\tau}, \boldsymbol{\xi}')\| \leq \beta(\|\boldsymbol{\xi} - \boldsymbol{\xi}'\|, \tau) + \gamma(\max_{1 \leq s \leq i-1} \|\delta\mathbf{u}_s\|)$. Given parameters $c_\gamma, c_\xi > 0$ we say that $\mathsf{a}$ is local-t-Iss at $\boldsymbol{\xi}_0$ if the above holds only for all $\boldsymbol{\xi}, \boldsymbol{\xi}', \delta\mathbf{u}_{1:\tau}$ such that $\|\boldsymbol{\xi} - \boldsymbol{\xi}_0\|, \|\boldsymbol{\xi}' - \boldsymbol{\xi}_0\| \leq c_\xi$ and $\max_t \|\delta\mathbf{u}_t\| \leq c_\gamma$.

Incremental stability implies that as the inital conditions $\|\boldsymbol{\xi} - \boldsymbol{\xi}'\| \to 0$ and $\max_{0 \leq s \leq i-1} \|\delta\mathbf{u}_t\| \to 0$, the trajectories induced by taking rolling out $\mathsf{a}$ from $\boldsymbol{\xi}$, and rolling out $\mathsf{a}$ from $\boldsymbol{\xi}'$ with additive input perturbations $\delta\mathbf{u}_{1:\tau}$ tend to zero in norm. This behavior needs only hold for initial conditions in a small neighborhood of a nominal state $\boldsymbol{\xi}_0$. Importantly, the perturbations $\delta\mathbf{u}_{1:\tau}$ are fixed pertubrations of inputs, applied to the *closed loop behavior* under the controllers. Our notion of incremental stability are similar too, but sublty different similar notions of past work. We provide an extended comparisons in Appendix E.2. Our main assumption is that the synthesis oracle described above produced primitive controllers which are consistent with, and incrementally stabilizing for, the demonstrated trajectories. Figure 1 demonstrates the effect of stabilizing primitive controllers.

**Assumption 3.1.** We assume that our synthesis oracle enjoys the following property. Let $\boldsymbol{\rho}_T = (\mathbf{x}_{1:T+1}, \mathbf{u}_{1:T}) \sim \mathcal{D}_{\exp}$, and let $\kappa_{1:T} = \mathtt{synth}(\boldsymbol{\rho}_T)$, partitioned into composite actions $\mathsf{a}_{1:H}$, with $\kappa_t(\mathbf{x}) = \bar{\mathbf{K}}_t(\mathbf{x} - \bar{\mathbf{x}}_t) + \bar{\mathbf{u}}_t$. We assume that, with probability one, $\kappa_{1:T}$ is consistent with $\boldsymbol{\rho}_T$[2], and that, for each $1 \leq h \leq H$, $\mathsf{a}_h = (\kappa_{t_h:t_h+\tau_{\mathrm{chunk}}-1})$ is local t-Iss at $\mathbf{x}_{t_h}$ with moduli $\gamma, \beta$ and parameters $c_\beta, c_\xi > 0$. We further assume that $\gamma$ and $\beta$ take the form

$$\gamma(u) = \bar{c}_\gamma \cdot u, \quad \beta(u, k) = \bar{c}_\beta e^{-(k-1)L_\beta} \cdot u, \quad \bar{c}_\gamma, \bar{c}_\beta > 0, \quad L_\beta \in (0, 1].$$

---

[1]Below, we recall definitions of classes of comparison functions in nonlinear control [38] as follows: we say a univariate function $\gamma : \mathbb{R}_{\geq 0} \to \mathbb{R}_{\geq 0}$ is *class* $\mathsf{K}$ if it is strictly increasing and satisfies $\gamma(0) = 0$. We say a bivariate function $\beta : \mathbb{R}_{\geq 0} \times \mathbb{Z}_{\geq 0} \to \mathbb{R}_{\geq 0}$ is *class* $\mathsf{KL}$ if $x \mapsto \beta(x, t)$ is class $\mathsf{K}$ for each $t \geq 0$, and $t \mapsto \beta(x, t)$ is nonincreasing in $t$.

[2]Note that this implies $\bar{\mathbf{x}}_t = \mathbf{x}_t$ and $\bar{\mathbf{u}}_t = \mathbf{u}_t$.

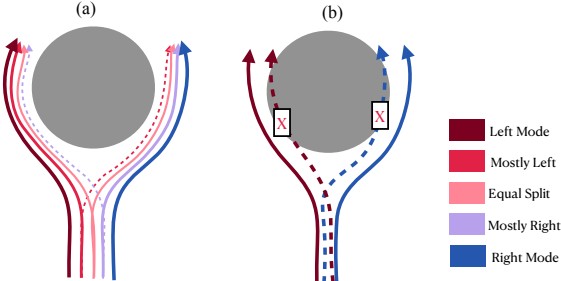

Figure 3: Graphical representation of total variation continuity (TVC) using the running "left mode/right mode" example. Panel *(a)* depicts a policy $\pi$ which is TVC, and thus interpolates between left and right modes probabilistically. Importantly, the TVC property applies to the distribution over composite actions, i.e. sequences of *primitive controllers*, as, in Definition 3.3; this ensures, for example, that following the left mode from slightly to the right of the obstacle (purple dotted line) still stabilizes to the idealized left mode trajectory (red). In panel *(b)*, we consider a policy which for TVC applies to the sequences of *raw control inputs* (which *is not* what occurs in Definition 3.3). This can lead to naive mode-switching that collides with the gray obstacle.

Lastly, we assume that for the expert trajectories and the primitive controllers drawn as above, it holds that satisfy $\max\{\|\mathbf{x}_t\|, \|\mathbf{u}_t\|\} \leq R_{\mathrm{dyn}}$ and $\|\bar{\mathbf{K}}_t\| \leq R_{\mathbf{K}}$ with probability one.

In Appendix K, we show that Assumption 3.1 holds whenever (a) the dynamics of our system are smooth (but not necessarily linear!) (b) the affine gains are chosen to stabilize the Jacobian linearizations of the system around the nominal trajectory.

**Definition 3.2** (Problem constants)**.** Throughout, we refer to constants $c_1, c_2, c_3, c_4, c_5 > 0$, which are polynomial in the terms in Assumption 3.1, and which are defined formally in Appendix K.

## 3.2 Simplified guarantees under total variation continuity

This section presents our main theoretical result: if one learns a chunking policy $\hat{\pi}$ that can compute the conditional distribution of composite actions at time steps given observation-chunks, then a stochastically smoothed version of this policy, $\hat{\pi}_\sigma$, has low imitation error. Define, for any length $\tau \in \mathbb{N}$, the *trajectory distance* between trajectories $\boldsymbol{\rho} = (\mathbf{x}_{1:\tau+1}, \mathbf{u}_{1:\tau})$, $\boldsymbol{\rho}' = (\mathbf{x}'_{1:\tau+1}, \mathbf{u}'_{1:\tau}) \in \mathscr{P}_\tau$

$$\mathsf{d}_{\mathrm{traj}}(\boldsymbol{\rho}, \boldsymbol{\rho}') := \max_{1 \leq k \leq \tau+1} \|\mathbf{x}_k - \mathbf{x}'_k\| \vee \max_{1 \leq k \leq \tau} \|\mathbf{u}_k - \mathbf{u}'_k\|. \tag{3.2}$$

In particular, we define $\mathsf{d}_{\mathrm{traj}}(\mathsf{o}_h, \mathsf{o}'_h)$ and $\mathsf{d}_{\mathrm{traj}}(\mathsf{s}_h, \mathsf{s}'_h)$ by viewing these as trajectories of length $\tau_{\mathrm{obs}} - 1$ and $\tau_{\mathrm{chunk}}$, respectively. Lastly, we define a per-timestep restriction of the expert distribution. In this section, we consider the case where the learner policy satisfies a total variation continuity (TVC) condition, defined below.

**Definition 3.3** (TVC of Chunking Policies)**.** We say that a chunking policy $\pi = (\pi_h)$ is total variation continuous with modulus $\gamma_{\mathrm{TVC}} : \mathbb{R}_{\geq 0} \to \mathbb{R}_{\geq 0}$, written $\gamma_{\mathrm{TVC}}$-TVC, if, for all $h \in [H]$ and any observation-chunks $\mathsf{o}_h, \mathsf{o}'_h \in \mathscr{P}_{\tau_{\mathrm{obs}}-1}$, $\mathsf{TV}(\pi_h(\mathsf{o}_h), \pi_h(\mathsf{o}'_h)) \leq \gamma_{\mathrm{TVC}}(\mathsf{d}_{\mathrm{traj}}(\mathsf{o}_h, \mathsf{o}'_h))$.

We depict the TVC property using our running left-right obstacle example in Figure 3. We stress that, in Definition 3.3, the TV bound on $\mathsf{TV}(\pi_h(\mathsf{o}_h), \pi_h(\mathsf{o}'_h))$ applies to the *composite actions* consisting of primitive controllers $\mathsf{a}_h = \kappa_{t_h:t_h+\tau_{\mathrm{chunk}}-1} \sim \pi_h(\mathsf{o}_h)$; it does not upper bound the TV distance between raw control inputs. Indeed, ensuring TVC of the latter can lead to the failure modes depicted in Figure 3(b). Next, we extract an expert "policy" from the expert demonstrations.

**Definition 3.4** (Expert "policy" with synthesized controllers)**.** For $h \in [H]$, we let $\mathcal{D}_{\mathrm{exp},h}$ denote the joint distribution of $(\mathsf{a}_h, \mathsf{o}_h)$, induced by drawing a trajectory $\boldsymbol{\rho}_T = (\mathbf{x}_{1:T+1}, \mathbf{u}_{1:T}) \sim \mathcal{D}_{\mathrm{exp}}$ from the expert distribution, $\kappa_{1:T} = \mathtt{synth}(\boldsymbol{\rho}_T)$ be the associated primitive controllers, letting $\mathsf{o}_h = (\mathbf{x}_{t_h-\tau_{\mathrm{obs}}+1:t_h}, \mathbf{u}_{t_h-\tau_{\mathrm{obs}}+1:t_h-1})$ be the associated observation-chunk at time $h$, and $\mathsf{a}_h = \kappa_{t_h:t_{h+1}-1}$ the associated composite action. We let $\pi_h^\star(\cdot) : \mathcal{O} \to \Delta(\mathcal{A})$ denote the condition distribution of $\mathsf{a}_h \mid \mathsf{o}_h$ under $\mathcal{D}_{\mathrm{exp},h}$.

The conditional distributions $\pi_h^\star(\cdot)$ are estimated when training a generative model to predict $\mathsf{a}_h$ from observations $\mathsf{o}_h$. Note that $\pi_h^\star(\cdot)$ (and $\mathcal{D}_{\mathrm{exp},h}$) is defined in terms of *both* expert demonstration

from $\mathcal{D}_{\mathrm{exp}}$ and the associated synthesized primitive controllers. In Lemma J.6, we show that when the synthesis oracle $\kappa_{1:T} = \mathtt{synth}(\boldsymbol{\rho}_T)$ produces primitive controllers consistent with the trajectories, than $\pi^\star = (\pi_h^\star)$ produces the same marginals over states as $\mathcal{D}_{\mathrm{exp}}$; that is, $\mathcal{L}_{\mathrm{marg},\varepsilon}(\pi^\star) = 0$.

**Theorem 1.** Suppose Assumption 3.1 holds, and suppose that $0 \leq \varepsilon < c_2$, and $\tau_{\mathrm{chunk}} \geq c_3$. Then, for any non-decreasing non-negative $\gamma_{\mathrm{TVC}}(\cdot)$ and $\gamma_{\mathrm{TVC}}$-TVC chunking policy $\hat{\pi}$, it holds that $\mathcal{L}_{\mathrm{marg},\varepsilon}(\hat{\pi}) \leq H\gamma_{\mathrm{TVC}}(\varepsilon) + \sum_{h=1}^{H} \mathbb{E}_{\mathsf{o}_h \sim \mathcal{D}_{\mathrm{exp},h}} \Delta_{(\varepsilon/c_1)} (\pi_h^\star(\mathsf{o}_h), \hat{\pi}_h(\mathsf{o}_h))$, which is at most $H\gamma_{\mathrm{TVC}}(\varepsilon) + \frac{c_1}{\varepsilon} \sum_{h=1}^{H} \mathbb{E}_{\mathsf{o}_h \sim \mathcal{D}_{\mathrm{exp},h}} [\mathcal{W}_{1,\mathsf{d}_{\mathrm{max}}} (\pi_h^\star(\mathsf{o}_h), \hat{\pi}_h(\mathsf{o}_h))]$.

The above result reduces the marginal imitation error of $\hat{\pi}$ to the sum over optimal transport errors between $\hat{\pi}$ and $\mathsf{a} \mid \mathsf{o}_h$ chosen by the expert demonstrators. Thus, if these are small, the local stabilization properties of the primitive controllers guaranteed by Assumption 3.1 ensure that errors compound at most linearly in problem horizon. The key ideas of the proof are given Appendix D, via a general template for imitation learning of general stochastic policies. This template is instantiated with a details in Appendix J.

### 3.3  A general guarantee via data noising.

To circumvent assuming that the learner's policy is TVC, we study estimating the conditionals under a popular data augmentation technique [36], where the learner is trained to imitate the conditional sequence of $\mathsf{a} \mid \tilde{\mathsf{o}}_h$, where $\tilde{\mathsf{o}}_h \sim \mathcal{N}(\mathsf{o}_h, \sigma^2 \mathbf{I})$ adds $\sigma^2$-variance Gaussian noise to the true observation-chunk. To understand this better, consider the following *smoothed* policy:

**Definition 3.5** (The smoothed policy). Let $\hat{\pi} = (\hat{\pi}_h)$ be a chunking policy. We define the *smoothed policy* $\hat{\pi}_\sigma = (\hat{\pi}_{\sigma,h})$ by letting $\hat{\pi}_{\sigma,h}(\cdot \mid \mathsf{o}_h)$ be distributed as $\hat{\pi}_h(\cdot \mid \tilde{\mathsf{o}}_h)$, where $\tilde{\mathsf{o}}_h \sim \mathcal{N}(\mathsf{o}_h, \sigma^2 \mathbf{I})$.

Appendix J.7.2 show's that Pinsker's inequality implies noising automatically enforces TVC This suggests that we can use some form of data noising to enforce the TVC property in Definition 3.3. Let's now consider a related problem: trying to estimate the optimal distribution over composite actions *conditioned on* a noised observation. This gives rise to a *deconvolution* of the expert policy, which can be thought as an inverse operation of data noising.

**Definition 3.6** (Noised Data Distribution and Deconvolution Policy). Let $\mathcal{D}_{\mathrm{exp},h}$ be as in Definition 3.4. Define $\mathcal{D}_{\mathrm{exp},\sigma,h}$ as the distribution over $(\tilde{\mathsf{o}}_h, \mathsf{a}_h)$ generated by $(\mathsf{o}_h, \mathsf{a}_h) \sim \mathcal{D}_{\mathrm{exp},h}$ and $\tilde{\mathsf{o}}_h \sim \mathcal{N}(\mathsf{o}_h, \sigma^2 \mathbf{I})$. We define the *deconvolution policy* $\pi_{\mathrm{dec},\sigma,h}^\star(\tilde{\mathsf{o}}_h)$ as the conditional distribution of $\mathsf{a}_h \mid \tilde{\mathsf{o}}_h$ under $\mathcal{D}_{\mathrm{exp},\sigma,h}$.

Analogously to $\pi^\star$, the policy $\pi_{\mathrm{dec},\sigma,h}^\star$ is what a generative model trained to generate $\mathsf{a}_h$ from noised observations $\tilde{\mathsf{o}}_h$ of $\mathsf{o}_h \sim \mathcal{D}_{\mathrm{exp}}$ learns to generate. Our next theorem states that, if our $\hat{\pi}$ approximates the idealized conditional distributation of composite actions given noised observations, then $\hat{\pi}_\sigma$, the smoothed policy, imitates the expert distribution with provable bounds on its imitation error:

**Theorem 2** (Reduction to conditional sampling under nosing). Suppose Assumption 3.1 holds. Let $c_1, \ldots, c_5 > 0$, defined in Definition 3.2, and let $\Theta_{\mathrm{Iss}}(x)$ denote a term which is upper and lower bounded by a $x$ times a polynomial in those constants and their inverses. Then, for $\varepsilon \leq \Theta_{\mathrm{Iss}}(1)$, if we choose $\sigma = \varepsilon/\Theta_{\mathrm{Iss}}(\sqrt{d_x} + \log(1/\varepsilon))$ and let $\tau_{\mathrm{chunk}} \leq c_3$ and $\tau_{\mathrm{chunk}} - \tau_{\mathrm{obs}} \geq \frac{1}{L_\beta} \log(c_1/\varepsilon)$,

$$\mathcal{L}_{\mathrm{marg},\varepsilon}(\hat{\pi}_\sigma) \leq \Theta_{\mathrm{Iss}} \left( \varepsilon H \sqrt{\tau_{\mathrm{obs}}} \cdot (\sqrt{d_x} + \log(\tfrac{1}{\varepsilon})) + \sum_{h=1}^{H} \mathbb{E}_{\tilde{\mathsf{o}}_h \sim \mathcal{D}_{\mathrm{exp},\sigma,h}} \left[ \Delta_{(\varepsilon^2)} \left( \pi_{\mathrm{dec},\sigma,h}^\star(\tilde{\mathsf{o}}_h), \hat{\pi}_h(\tilde{\mathsf{o}}_h) \right) \right] \right),$$

which is upper bounded by at most $\Theta_{\mathrm{Iss}} \left( \varepsilon H \sqrt{\tau_{\mathrm{obs}}} \cdot (\sqrt{d_x} + \log(1/\varepsilon)) + \frac{1}{\varepsilon^2} \sum_{h=1}^{H} \mathbb{E}_{\mathsf{o}_h \sim \mathcal{D}_{\mathrm{exp},\sigma,h}} \left[ \mathcal{W}_{1,\mathsf{d}_{\mathrm{max}}} \left( \pi_{\mathrm{dec},\sigma,h}^\star(\tilde{\mathsf{o}}_h), \hat{\pi}_h(\tilde{\mathsf{o}}_h) \right) \right] \right)$.

To reiterate, Theorem 2 guarantees imitation of the distribution of marginals and final states of $\mathcal{D}_{\mathrm{exp}}$ by replacing the explicit TVC assumption with noising, and the resulting guarantee applies to the *smoothed policy* $\hat{\pi}_\sigma$ which adds smoothing noise back in. Appendix J gives a number of additional results. In Appendix I, we show that the proof framework, outlined in Appendix D, which under lies the proofs of Theorems 1 and 2, is essentially sharp in the worst case. Moreover, in Appendix C.3, we discuss the merits and drawbacks of our use of the synthesis oracle, and how it circumvents some of the challenges encountered in behavior cloning in past work. The key intuition behind the proof of Theorem 2 is depicted in Figure 4, and full proof sketch is deferred to Appendix C.2

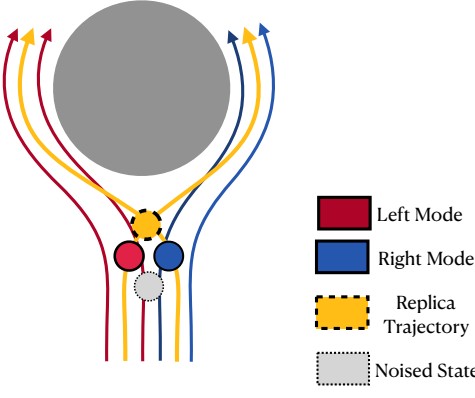

Figure 4: Multi-modal demonstrations traverse an obstacle left or right, exhibiting a pure bifurcation. We consider perturbing expert data on the right mode (blue circle) to a noised datum (gray circle). We show that generative behavior cloners learn to deconvolve this noise, creating a virtual "replica" sample (red circle) following the left mode, such that the replica and original are i.i.d. given the noised one. When the red circle's primitive controllers are rolled from from the blue circle, this leads to a trajectory (yellow circle) which interpolates across the bifurcations. Marginalizing over this process, the yellow trajectories probabilistically interpolate between red and blue modes, and (approximately) match the per-time-step marginal over expert distributions.

**Left Mode**

**Right Mode**

**Replica Trajectory**

**Noised State**

## 4  HINT: Instantiating Data Noising with DDPMs

We now instantiate Theorem 2 by showing that one can learn a policy $\hat{\pi}$ for which the error terms in Theorems 1 and 2 are small by fitting a DDPM to noise-smoothed data.    Our proposed algo-

---

**Algorithm 1** **H**ierarchical **I**mitation via **N**oising at Inference **T**ime (HINT)

[h]
 1: **Initialize** Synthesis oracle `synth`, sample sizes $N_{\mathrm{exp}}, N_{\mathrm{aug}} \in \mathbb{N}$, $\sigma \geq 0$, DDPM step size $\alpha > 0$, DDPM horizon $J$, function class $\{\mathbf{s}_\theta\}_{\theta \in \Theta}$, gain magnitude $R > 0$, empty data buffer $\mathfrak{D} \leftarrow \emptyset$.
     % For no smoothing, set $\sigma = 0$ and $N_{\mathrm{aug}} = 1$
 2: **for** $n = 1, 2, \ldots N_{\mathrm{exp}}$ **do**
 3:     Sample $\boldsymbol{\rho}_T = (x_{1:T+1}, u_{1:T}) \sim \mathcal{D}_{\mathrm{exp}}$ and set $\kappa_{1:T} = \mathtt{synth}(\boldsymbol{\rho})$
         % Segment $\mathbf{o}_{1:H}$ from $\boldsymbol{\rho}_T$ and $\mathbf{a}_{1:H}$ from $\kappa_{1:T}$
 4:     **for** $i = 1, 2, \ldots, N_{\mathrm{aug}}$ and $h = 1, 2, \ldots, H$ **do**
 5:         Sample $\tilde{\mathbf{o}}_h \sim \mathcal{N}(\mathbf{o}_h, \sigma^2 \mathbf{I})$, $j_h \sim \mathrm{Unif}([J])$ and $\boldsymbol{\gamma}_h \sim \mathcal{N}(0, (j_h \alpha)^2 \mathbf{I})$.
 6:         $\mathfrak{D} \leftarrow \mathfrak{D}.\mathrm{append}\left(\{(\mathbf{a}_h, \tilde{\mathbf{o}}_h, j_h, \boldsymbol{\gamma}_h, h)\}\right)$
 7: Fit $\theta \in \arg\min_{\theta \in \Theta} \mathcal{L}_{\mathrm{DDPM}}(\theta, \mathfrak{D})$, and let $\hat{\pi} = (\hat{\pi}_h)$ be given by $\hat{\pi}(\cdot \mid \mathbf{o}_h) = \mathtt{DDPM}(\mathbf{s}_{\theta,h}, \mathbf{o}_h)$.
 8: **return** $\hat{\pi}_\sigma = (\hat{\pi}_{\sigma,h})$, by smoothing $\hat{\pi}$ as per Definition 3.5.

---

rithm, HINT (Algorithm 1) combines DDPM-learning of chunked policies as in [19] with a popular form of data-augmentation [36]. We collect $N_{\mathrm{exp}}$ expert trajectories, synthesize gains, and segment trajectories into observation-chunks $\mathbf{o}_h$ and composite actions $\mathbf{a}_h$ as described in Section 2. We perturb each $\mathbf{o}_h$ to form $N_{\mathrm{aug}}$ chunks $\tilde{\mathbf{o}}_h$, as well as horizon indices $j \in [J]$ and inference noises $\boldsymbol{\gamma} \sim \mathcal{N}(0, (\alpha j_h)^2 \mathbf{I})$, and add these tuples $(\mathbf{a}_h, \tilde{\mathbf{o}}_h, j_h, \boldsymbol{\gamma}_h, h)$ to our data $\mathfrak{D}$. We end the training phase by minimizing the standard DDPM loss [62]:

$$\mathcal{L}_{\mathrm{DDPM}}(\theta, \mathfrak{D}) = \sum_{(\mathbf{a}_h, \tilde{\mathbf{s}}_h, j_h, \boldsymbol{\gamma}_h, h) \in \mathfrak{D}} \left\| \boldsymbol{\gamma}_h - \mathbf{s}_{\theta,h}\left(e^{-\alpha j} \mathbf{a}_h + \sqrt{1 - e^{-2\alpha j}} \boldsymbol{\gamma}_h, \tilde{\mathbf{o}}_h, j_h\right) \right\|^2. \quad (4.1)$$

Our algorithm differs subtly from past work in Line 8: motivated by Theorem 2, we add smoothing noise *back in* at test time. Here, the notation $\mathtt{DDPM}(\mathbf{s}_{\theta,h}, \cdot) \circ \mathcal{N}(\mathbf{o}_h, \sigma^2 \mathbf{I})$ means, given $\mathbf{o}_h$, we perturb it to $\tilde{\mathbf{o}}_h \sim \mathcal{N}(\mathbf{o}_h, \sigma^2 \mathbf{I})$, and sample $\mathbf{a}_h \sim \mathtt{DDPM}(\mathbf{s}_{\theta,h}, \tilde{\mathbf{o}}_h)$. We now state an informal guarantee for HINT, deferring a formal statement to Appendix C.5.

**Theorem** (Informal Theorem). Suppose that the system dynamics are smooth and that Assumption 3.1 holds for the linearized system. Then there is a choice of the parameters in HINT that is polynomial in all problem parameters such that for $N_{\mathrm{exp}}$, polynomially large in problem parameters, $\mathcal{L}_{\mathrm{marg},\varepsilon}(\hat{\pi}_\sigma) \leq \Theta\left(\varepsilon H \sqrt{\tau_{\mathrm{obs}}}(\sqrt{d_x} + \log(1/\varepsilon))\right)$ with high probability.

### 4.1  Experimental Results

In this section, we demonstrate the benefits of diffusing low level controllers, and of our approach to data noising. We explain the environments in greater detail, along with all training and compu-

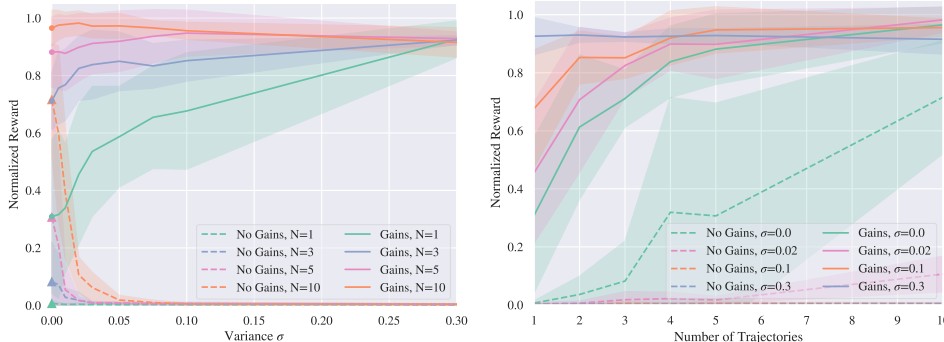

Figure 5: Performance of diffusing chunks of actions $\bar{\mathbf{u}}_{t_{h-1}:t_h-1}$ ("No Gains") versus jointly diffusing actions $u_{t_{h-1}:t_h-1}$, reference states $\bar{\mathbf{x}}_{t_{h-1}:t_h-1}$ and gains $\mathbf{K}_{t_{h-1}:t_h-1}$ for a 2-D quadrotor system with thrust-and-torque-based control. Different noise levels $\sigma$ and number of trajectories $N$ are shown. Mean and standard deviation are shown across 5 training seeds.

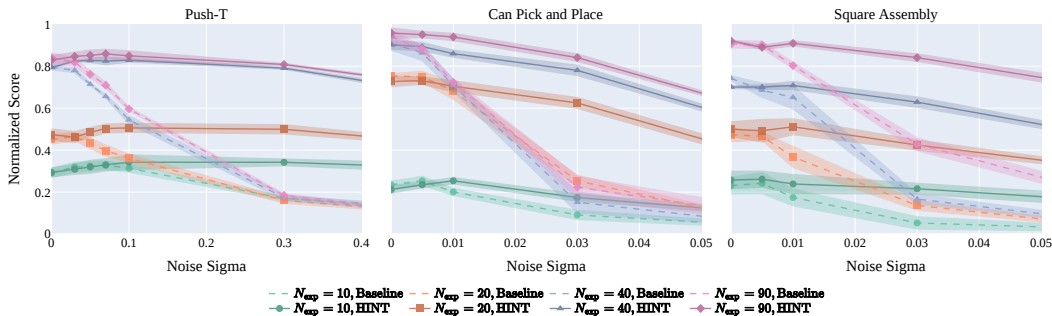

Figure 6: Performance of baseline $\hat{\pi}$ and noise-injected $\hat{\pi} \circ \mathsf{W}_\sigma$ HINT policy for different $\sigma$. We use 4 training seeds with 50 and 22 test trajectories per seed for PushT and Can and Square Environments respectively. Mean and standard deviation of the test performance on the 3 best checkpoints across the 4 seeds are plotted. The $\sigma$ values correspond to noise in the normalized $[-1, 1]$ range.

tational details in Appendix O. [3] Figure 5 compares the performance of diffusing (chunks of) raw control inputs to diffusing (chunks of) gain matrices for a canonical model of a 2-d quadrotor. We find that diffusing gain matrices yields dramatic improvements in performance, in particular allowing a *single imitated trajectory* to outperform learning raw control inputs from 10 demonstrations.

Next, empirically evaluate the effect on policy performance of our proposal to inject noise back into the dynamics at inference time. We consider three challenging robotic manipulation tasks studied in prior work: PushT block-pushing [19]; Robomimic Can Pick-and-Place and Square Nut Assembly [42] (we direct the reader to Chi et al. [19] for an extensive empirical investigation into the performance of diffusion policies in the un-noised $\sigma = 0$ regime). We display the results of our experiments in Figure 6. Observe that the performance degredation of the replica policy from the unnoised $\sigma = 0$ variant is minimal across all environments and even leads to a slight but noticeable improvement in the small-noise regime for PushT (and low-data Can Pick and Place). In the presence of non-negligible noise HINT significantly outperforms the conventional policy $\hat{\pi}$ (obtained by noising observations at training but not test time), as predicted by our theory.

## 5  Discussion

This work considerably loosened assumptions placed on the *expert distribution* by introducing a synthesis oracle responsible for stabilization. How best to achieve low-level stabilization remains an open question. We hope that this work encourages further empirical research into improving the stability of imitation learning, either via the hierarchical route proposed in this paper or via new innovations.

---

[3]Code for PushT, Robomimic experiments can be found at https://github.com/pfrommerd/diffusion_policy_pt. Quadrotor experiments are in https://github.com/pfrommerd/stanza/

## Acknowledgments

We would like to thank Cheng Chi for his extensive help in running the code in [19] and Benjamin Burchfiel for numerous insightful discussions and intuitions about DDPM policies. Lastly, we thank Alexandre Amice for his helpful feedback on an earlier draft of this manuscript. AB acknowledges support from the National Science Foundation Graduate Research Fellowship under Grant No. 1122374 as well as the support of DOE through award DE-SC0022199. MS and RT acknowledge support from Amazon.com Services LLC grant; PO# 2D-06310236. DP and AJ acknowledge support from the Office of Naval Research under ONR grant N00014-23-1-2299.

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
