# OpenReview forum: "Provable Guarantees for Generative Behavior Cloning: Bridging Low-Level Stability and High-Level Behavior"
_NeurIPS.cc/2023/Conference — NeurIPS 2023 poster_

### Official Review · Reviewer_dAio · 2023-06-29

**Soundness:** 3 good
**Presentation:** 2 fair
**Contribution:** 3 good
**Rating:** 6
**Confidence:** 3

**Summary:**

The paper presents a simple hierarchical approach for non-Markovian imitation learning along with a rigorous theoretical analysis.
The method trains a high level policy, given by a Denoising Diffusion Probabilistic Model (DDPM), to predict a sequence of linear control gains based on a state-action history.
The DDPM is trained in supervised fashion (using the standard DDPM loss) where the labels (the controller gains) are computed from the given expert trajectories using an oracle (e.g. based on linearizations along the trajectory). To address covariate shift (compounding errors) in imitation learning the method applies a form of data augmentation by adding noise to the state-action history. In contrast to prior work, these perturbations are also applied during inference, which is theoretically motivated and empirically shown to improve imitation performance.
The main contributions seems to be the thorough analysis of this method, which leads to Theorem 1, a bound on the imitation performance.
The applicability of the proposed method is demonstrated on simulated robot tasks, although no comparisons with prior work are provided.

**Strengths:**

Novelty & Relevance
-------------------
Both the method and in particular the analysis seem to be novel and non-Markovian imitation learning is an important field of research. I would not expect the proposed method to outperform other (theoretically less well understood) methods, but I do think that the general approach of using a diffusion model as a high-level policy to predict controller gains could be used by future work that is more focused on practical performance. Furthermore, the theoretical analysis could be valuable for theorists in the field of imitation learning. However, I am not very familiar with related theoretical work and cannot assess well, which aspects/techniques of the analysis might be most relevant.

Soundness
---------
The algorithm seems sounds. I guess one could argue that it might be simpler and more stable if the higher-level policy would predict a trajectory segment instead of the gains, given that we assume access to an oracle that can provide us with stabilizing gains. However, querying the oracle could be too costly, so directly predicting the gains can be sensible.

I cannot fully confirm the accuracy of the theoretical results because verifying the complete derivations would take significant effort. However, the claims (including Theorem 1) seem reasonable, apart from non-critical issues (see Questions below).

Clarity
-------
The paper is well-written. The paper is very strict in terms of notation, definitions, and statements, which increases clarity by reducing ambiguities.

**Weaknesses:**

Clarity
-------
Although I mentioned the rigor as a strength, it also feels a bit pedantic. Furthermore, it is cumbersome to keep track of the different constants and definitions making the paper hard to read and making it easy to get lost in details.

Evaluation
----------
Despite the fact that the contributions are mainly in the theoretical analysis, quantitative comparisons with alternate methods that have been used in these environments would be useful. I don't know which prior work was used in that environment that also provided code, but I assume it should be possible to find suitable baselines that can be easily tested.



**Questions:**

Line 216 states that "each term in (3.2) can be made arbitrarily small by decreasing the amount of noise $\sigma$ in the augmentation [...]". Isn't the first term anti-proportional to the noise?

Line 274: "all sequences $s'\_1 = s\_1$ and  $s'\_{h+1} = s\_{h+1}$ [...]". I'm wondering if this should be "all sequences $s'\_h = s\_h$ and  $s'\_{h+1} = s\_{h+1}$ [...]"!?

Line 325: "We let the expert policy $\pi^*$ be the concatenation of policies $\pi\_h^*$ [...]". Is this an additional assumption? Clearly, the expert will in general not apply this particular form of a hierarchical policy.

**Limitations:**

I think that the limitations are sufficiently clear.

---

> ### Author Rebuttal · Authors · 2023-08-09
>
> Thank you for your feedback.
> # Clarity
> Regarding the issue of clarity, we will clean up the notation and provide greater intuition for the mathematical abstractions in the revision.  For example, we have now replaced $\circlearrowleft$ symbol with $\mathrm{rep}$ for replica-quantities.
>
> # Empirical Evaluation
> On the subject of empirical evaluation, note that our paper’s primary contribution is in the novel mathematical contributions that allow us to prove the first general imitation learning guarantees in regimes where the demonstrator policies are arbitrarily unstable.  The empirical section is intended to demonstrate that our proposed intervention (adding noise at inference time) is beneficial with respect to the benchmark of standard practice, which is adding noise only at training time but not at inference time. For the revision, we will also include experiments on 2D quadrotor – a system known to be unstable – to demonstrate the benefits of synthesizing gains.
>
> There has been a lot of excellent empirical work that has demonstrated the advantages of diffusion-based policies over prior approaches. We refer the reader to [1] for further exposition.
>
> # Questions
> We will now address the questions:
> 1. Thank you for pointing this out, we will make this more clear in the revision.  What we mean to say is that as the number of demonstrators grow, we may take epsilon arbitrarily small and thus, by tuning the noise sigma appropriately, our proposed algorithm can achieve arbitrarily small loss with sufficiently many demonstrations.
> 2. To clarify, we mean all sequences $(s_{1:H+1}',a_{1:H}')$ whose initial condition is $s_1' = s_1$, and whose dynamics obey $s_{h+1}' = F_h(s_h',a_h')$.
> 3. The crucial part of the stability is that it is *multi-step*, so that the differences between actions $a’_h$ and $a_h$ can accumulate over time. Due to space constraints, we opted for the compact definition but can add more clarification in the revision.
> 4. This is not an extra assumption, but rather a definition.  Note that while we colloquially refer to policies, formally we consider a distribution on trajectories.  We then rigorously *define* a policy at each time step $h$ to be the conditional distribution of an action given the history up to time $h$.  Thus a global policy (defined on all time steps) is just the concatenation of each of the policies at each time step.  This is equivalent to factoring a joint distribution as a product of conditional distributions.  We will better clarify this in the revision.
>
> [1] C. Chi, S. Feng, Y. Du, Z. Xu, E. Cousineau, B. Burchﬁel, and S. Song. Diffusion policy:Visuomotor policy learning via action diffusion. arXiv preprint arXiv:2303.04137, 2023.

---

> > ### Comment · Reviewer_dAio · 2023-08-18
> > **Thank you for the clarifications**
> >
> > Thank you for the clarifications. I maintain my rather positive assessment.

---

### Official Review · Reviewer_rprV · 2023-07-04

**Soundness:** 2 fair
**Presentation:** 3 good
**Contribution:** 2 fair
**Rating:** 3
**Confidence:** 3

**Summary:**

This paper proposes a theoretical framework for studying the imitation of expert demonstrations in nonlinear dynamical systems. The framework guarantees accurate imitation of expert trajectories by invoking low-level controllers and a stochastic continuity property of the learned policy. The authors also introduce an algorithmic trick for ensuring total variation continuity, which combines a popular data-augmentation regimen with augmentation noise at execution time. The paper includes a detailed proof of the framework and empirical validation of the proposed augmentation strategy in simulated robotic manipulation tasks

**Strengths:**

1. The paper proposes a novel theoretical framework for studying the imitation of expert demonstrations in nonlinear dynamical systems.
2. The framework guarantees accurate imitation of expert trajectories by invoking low-level controllers and a stochastic continuity property of the learned policy.
3. The authors introduce a novel algorithmic trick for ensuring total variation continuity, which combines a popular data-augmentation regimen with augmentation noise at execution time.
4. The paper includes a detailed proof of the framework and establishes stability guarantees for sequences of primitive controllers in non-linear control systems.
5. The proposed augmentation strategy is empirically validated in simulated robotic manipulation tasks.
6. The paper provides a comprehensive review of related work in the field of imitation learning and compares the proposed framework to past work in the appendix.

**Weaknesses:**

1. The paper relies on either synthesized primitive controllers or low-level stabilizing controllers built into problem environments to take advantage of local stability. Developing a more comprehensive approach to stability (perhaps one that does not require explicit gain synthesis and extends to non-smooth systems) is an exciting direction for future work.
2. The paper focuses on simulated robotic manipulation tasks, and it is unclear how well the proposed framework would generalize to other domains or real-world scenarios.
3. The paper does not provide a detailed analysis of the computational complexity of the proposed algorithmic trick for ensuring total variation continuity.
4. The paper does not compare the proposed framework to state-of-the-art methods in the field of imitation learning on benchmark datasets.

**Questions:**

1. I think it is still important to comprehensive study of this method. What’s the influence of hyperparameters to this method? Should there be any ablation study included in this paper?
2. What’s the computational complexity of the proposed algorithmic trick for ensuring total variation continuity?

**Limitations:**

1. The paper assumes that the low-level controllers are either learned or implicit in position-command control, which may not always be the case in practice.
2. The paper does not provide a detailed analysis of the effect of hyperparameters on the performance of the proposed framework.
3. The paper does not consider the effect of imperfect state estimation or perception on the accuracy of the imitation policy.
4. The paper does not address the issue of exploration in the imitation learning setting, which is an important problem in reinforcement learning.

---

> ### Author Rebuttal · Authors · 2023-08-09
>
>
> Thank you for the feedback.  We will address each point below.
>
> # Weaknesses
>
> 1.The main technical contribution of the paper is formalizing a notion of hierarchical stability that is flexible enough to accommodate multi-modality. Note that our analysis extends to any synthesis oracle that provides increment lly stable primitive controllers, significantly generalizing linear gains. This generalization will be included in the revision. Such more general synthesis oracles can be implemented from reinforcement learning interactions with a simulator of the environment.
>
> 2.While the simulation-to-real problem is well-known and well-documented, it is standard to first understand an algorithm’s benefits in simulation before moving to hardware.  While our paper focused mainly on theoretical advances, we included some simulation experiments for the sake of concreteness, as well as to demonstrate the practicality of our algorithm.
>
> 3. The data augmentation is simply adding Gaussian noise to observations, so its computational complexity is negligible compared to model training.  We will emphasize this in the revision. Notice that we never explicitly constructed the replica policy in our analysis, it is merely a tool to describe the idealized behavior of a policy noise at test and inference time in the limit of large sample sizes.
>
> 4. Our work is primarily about providing novel theoretical insights into the use of generative-modeling based policies for imitation learning. We provide the first rigorous mathematical guarantees for imitating behaviors, far surpassing the generality of past work. While we make numerous algorithmic suggestions, including noising-at-inference and synthesizing low level controllers, the goal of the work is first and foremost mathematical.
>
> # Questions
>
> 1. Because the primary algorithmic modification is to add noise at inference time, as opposed to only adding the augmentation at train time, this was the benchmark to which we compared our algorithm empirically.  There are many empirical studies of diffusion models applied to similar tasks.  We also conducted a dataset ablation by subsampling the number of expert trajectories trained.
>
> 2. The addition of gaussian noise at inference time ensures total variation continuity (TVC). Thus, we do not need to alter the weights of the networks to ensure TVC (and thus the computational overhead is negligible). Indeed, we prove in the appendix that for any function $f(x)$, the random function $x \mapsto f(x + \sigma w)$, where $w \sim \mathcal{N}(0,I_d)$ is $O(\sigma)$-total variation continuous, *regardless of the function* $f$
>
>
> # Limitations
>
> 1. We agree access to low-level controllers is a limitation. As noted in the introduction, pure supervised learning is not sufficient for provable guarantees on imitation learning. Past works have required interactions with the environment (DAGGER [1], DART [2]), or imitating a much simpler, stabilizing behaviors  with gradient/adversarial queries (TaSIL [3]). We present the first result without adaptive data collection for general behaviors. We show that we replace the stabilizing expert from TaSIL [3] with an oracle that synthesizes stabilizing primitive controllers, which stabilize only locally, which is much weaker. While this does not include all practical situations, we can synthesize these gains whenever the dynamics are smooth and we have a good differentiable simulator. We are working with collaborators who use hydroelastic simulation for whole-body manipulation to test out our method.
>
> 2. See (1) in Question; moreover, as is common in theoretical works, we do not focus on hyperparameter optimization. We do, however, show the importance of the smoothing parameter sigma in our experiments.
>
> 3. As we remark, because visuomotor experiments have been done in similar settings in many other works (cited in the paper), we consider direct state access for our experiments for convenience.  Note that our theory does allow for much greater generality.
>
> 4. While exploration is certainly important in reinforcement learning, this paper is concerned with the topic of purely offline imitation learning, a paradigm that has seen increased recent interest due to the fact that it does not require the often unrealistic assumption of online interaction that defines online RL.
>
> [1] Ross, Stéphane, Geoffrey Gordon, and Drew Bagnell. "A reduction of imitation learning and structured prediction to no-regret online learning." Proceedings of the fourteenth international conference on artificial intelligence and statistics. JMLR Workshop and Conference Proceedings, 2011.
>
> [2] Laskey, Michael, et al. "Dart: Noise injection for robust imitation learning." Conference on robot learning. PMLR, 2017.
>
> [3] Pfrommer, Daniel, et al. "Tasil: Taylor series imitation learning." Advances in Neural Information Processing Systems 35 (2022): 20162-20174.

---

### Official Review · Reviewer_VsMF · 2023-07-07

**Soundness:** 2 fair
**Presentation:** 2 fair
**Contribution:** 2 fair
**Rating:** 6
**Confidence:** 1

**Summary:**

The paper introduces an imitation learning framework for handling multimodal and non-Markovian demonstrations. This is done by using a hierarchy where rather than predicting low-level controls directly, the learner predicts sequences of primitive controllers. These controllers are time-varying affine mappings from states to control inputs. The authors show that this approach allows us to reformulate our task as imitation in a composite MDP, where composite states and actions refer to subsequences of trajectories and primitive controllers respectively. To handle multimodality, they use diffusion models (DDPM) to parameterize the learner. The primary contribution of the paper is to bound the loss of the proposed approach (TODA), relying primarily on the stability of the lower-level controllers. Some empirical results are demonstrated on some robot manipulation tasks, where the proposed algorithm outperforms the baseline policy.

**Strengths:**

The high-level motivation for the approach is sound and the practical algorithm seems to perform well.

The paper is comprehensive and filled with technical content and derivations. I went through the analysis in the main body of the paper and did not find any glaring errors, although certain parts were hard to parse (see below).

**Weaknesses:**

Honestly I have a hard time following Section 4. More detailed questions are below. Given that the crux of the contribution lies in this theoretical analysis, I'm more than happy to adjust my score post-rebuttal if I can get some things cleared up.

Training a diffusion model to use low-level controllers as primitives as not new [1].

Would have preferred a more comprehensive experimental evaluation with competitive baselines, but given the theoretical contribution I suppose this isn't fully necessary.

The proposed approach heavily relies on sufficiently smooth dynamics and the existence of an oracle which can synthesize primitive controllers given trajectories. This is fairly restrictive and prevents this work from being applicable to many existing control/robotics applications. Given that the contribution is primarily theoretical, this isn't a major issue for me.

[1] Ajay, Anurag, et al. "Is conditional generative modeling all you need for decision-making?." arXiv preprint arXiv:2211.15657 (2022).

**Questions:**

What is $d_{\mathcal{A}}$? Not the formal definition, but conceptually what is it?

What is the purpose of the replica policy? I'm not sure I understand how this part of the proof functions.

Is the smoothing kernel $W_{\sigma}$ the same as the data augmentation function?

I don't understand L304-307.

Typos:

L52, "... which is both both ..."

L60, "Wassertstein-like"

L106, "... we assume that (2.1) deterministic ..."

L342, "subtley"

===
I have read the rebuttal and bumped my score up slightly.

**Limitations:**

Yes

---

> ### Author Rebuttal · Authors · 2023-08-09
>
> # Novelty and Theoretical Contributions
>
> Thanks for your feedback. We agree that training a diffusion model with low-level primitives is not a new idea [1], and (as noted in our paper) in most works that use diffusion, they apply position control rather than torque control which can also be viewed as primitive. The purpose of this paper is to give a rigorous analysis of this approach, and to re-formulate what *stability* means in this hierarchical context (which is not found in the theoretical literature). We provide a rigorous framework and analysis to that end. We provide a complementary way to obtain primitives - namely through linear gains from Riccati equations. By contrast, [1] requires an inverse dynamics model (which may be challenging non one-step controllable systems).
>
> # Future Experiments
>
> While the focus of this paper is mainly theoretical, in the revision, we will also include a quadrotor experiment with synthesized gains to showcase our method. We are also working with collaborators to apply the method (with synthesized gains) to a hydroelastic simulator of a soft whole-body robot manipulator. Notice that in this simulator, as in *any differentiable simulator*, *we can synthesize gains*, so that the method is practically feasible. While addressing non-smooth systems is an interesting algorithmic direction for future work, the  assumptions {smoothness, Jacobian stabilizability} can be replaced by incremental stability of the primitive controllers (which these assumptions yield for linear primitive controllers as a special case). We will expand on this point in the revision.
>
>
> # Action Distance:
>
> Recalling that a composite action is just a sequence of $\tau_c$ primitive controllers with parameters in Euclidean space,
> intuitively $d_A$ is the maximal Euclidean distance between the parameters of the primitive controller on each of the time steps.  In order to provide greater generality to locally stable systems, we add a term that is infinite for when two sequences of primitive controllers are too far apart for us to be able to say anything about them.  Intuitively, one can ignore the indicator function if you condition each step of the proof (and the results) on the event that actions and induced trajectory are sufficiently close together for local stability to hold.
>
> # Replica Policy
>
> The *replica policy* is a fictitious policy that we use in our analysis. It represents what would happen if one
>
> (a) perfectly trained a generative model to predict  $a_h \mid s_h + w_h$ (actions given smoothed states), and
>
> (b) selected actions at state x and step h by applying $\mathrm{trained}~\mathrm{model}(s_h + w_h)$.
>
> By contrast, the *deconvolution policy* is trained as in (a) but does not use smoothing noise as in (b). The name ``replica’’ comes from the fact that we can view the process of smoothing and then deconvolving a state as creating a *replica* of that state, which is close to it, and has the same marginal distribution (this is how it’s used in physics).   The key insights are that (1) training a policy with data augmentation and then inference-time noise injection converges to the replica policy and (2) unlike the deconvolution policy, the *replica policy*, is "close’’ to the $\pi^\star$ policy in many ways which are important for the analysis. We did our best to explain this in the text.
>
> For example, if $s_h ~ P^\star_h$, the marginal distribution over actions under the replica policy and the $\pi^\star$ are the same. This means that, following the replica policy, we essentially see similar states that we did to follow pi^*. Thus, we don’t suffer the problem of finding states that are out of the distribution of the demonstrator. By avoiding this distribution shift, our analysis shows that our learned policy imitates the replica policy. Because the replica policy is close to $\pi^\star$, argue that imitating the replica policy (in joint distribution) suffices to imitate $\pi^\star$ in terms of state marginals.   The true policy $\pi_h^\star$ maps state $x$ -> distribution over action $a$, and the *deconvolution policy* maps $x$ -> the distribution of $\pi_h^\star(x) | x_h+w_h = x$, under $x_h ~ P^\star_h$, then the *replica policy* maps $x \mapsto x+w$ to a deconvolution policy evaluated at $x+w$. One way to think of this is saying that $x$ is smoothed to a point $x’$, and deconvolved to another point $x’’$.
>
> The crucial property is that, if $x\sim P^\star_h$, then
>  $(x,x’’ \mid x)$ are i.i.d random variables.
> Yes the smoothing kernel is the same as the data augmentation; our proof holds in greater generality than simple gaussian noise and so we use a different symbol to emphasize this generality. In the revision, we apply consistent notation to avoid this confusion
> See the explanation on the replica policy.
>
> # Typos
> Thank you for finding the typos; these will be corrected in the revision.
>
>
> [1] Ajay, Anurag, et al. "Is conditional generative modeling all you need for decision-making?." arXiv preprint arXiv:2211.15657 (2022).

---

> > ### Comment · Reviewer_VsMF · 2023-08-12
> >
> > I appreciate the clarification from the authors and look forward to seeing the quadrotor experiments. I am still mostly in favor of acceptance and adjusted my score upwards slightly.

---

### Official Review · Reviewer_RM4v · 2023-07-07

**Soundness:** 3 good
**Presentation:** 3 good
**Contribution:** 3 good
**Rating:** 7
**Confidence:** 3

**Summary:**

This is a theoretical paper that studies imitation learning from demonstrations that exhibit multi-modality. This method separate the demonstrations into chunks, then uses diffusion models to learn a policy conditioned on the memory of hindsight prefix of the trajectory to output an affine time-varying controller. Gaussian noises are added to the demonstrations for data augmentation, and in the learned controllers' outputs to enhance performance.

**Strengths:**

Originality: this paper is original in proposing an algorithm to learn hierarchical policy by leveraging diffusion models and injected noises from potentially non-markovian expert demonstrations.
Quality: this paper has strong theoretical support.
Clarity: the algorithms and theories are clearly explained.
Significance: the paper provides a new insight to solving non-markovian imitation learning problems by combining trajectory optimization tools, distribution learning in a hierarhical manner.

**Weaknesses:**

In the experiment, three benchmarks are utilized. However, it is not mentioned in the maintext nor in the appendix how non-markovian and multi-modal the demonstrations are. As a result, the validity of the proposed algorithm is less convincible despite all the theoretical guarantees.
Also, the symbols in this paper are too complicated to parse. There seem to be some symbols misusage in some of the formulas, making those formulas confusing. Please refer to the question section for the those potential errors.

**Questions:**

1. In line 5 of Algorithm 1, shouldn't the mean of \gamma_h be 0 instead of a_h? In Eq.3.1, \gamma_h is sampled from D to compose the a DDPM loss function of which the update rule matches Eq 2.2. Then, \gamma_h corresponds to the \gamma in line 155, which is a standard Gaussian.
2. If \gamma_h indeed has its mean equal to a_h, how to understand the sum in Eq3.1? Why adding such noised action to e^{-\alpha j} a_h?
3. If \gamma_h has its mean equal to 0, why matching s_\theta,h(...) with \gamma_h, a Gaussian noise?
4. What do ...\vee... \leq max ... mean in line 275? Applying \vee to real values?

**Limitations:**

The author mentioned the limitation of the proposed approach in the paper. The limitation, however, does not undermine the quality and contribution of this paper.

---

> ### Author Rebuttal · Authors · 2023-08-09
>
> Thank you for the positive review and helpful feedback. One of our benchmarks, Push-T was proposed by [1] as an explicitly multi-modal task; Figure 4 in [1] illustrates that one can go left or right around the T-object. We will be sure to include a reference to this figure in our revision.
>
> Though the primary prupose of our paper is theoretical in nature, we are currently working on a 2D quadrotor environment as well. In our preliminary findings, the replica policy consistently outperforms the deconvolution policy. This suggests that adding noise at inference time does indeed improve performance. We suspect to see the benefits of adding noise once we also finish our implementation of diffusing feedback gains, and with a Wasserstein measure of discrepancy to capture the fact, shown in our analysis, that the replica policy matches the expert distribution on marginals at each time step but not, necessarily, on the joint trajectory distribution. We will include all these additional experiments in the final version.
>
> We apologize for the notational overhead. We replaced the $\circlearrowleft$ symbol with $\mathrm{rep}$ for replica-quantities in the revision. We made sure to use consistent subscript-types and letter names for all quantities - W is a kernel, P a distribution, subscript-$\mathrm{dec}$ is for deconvolution, $\pi$ is a policy, and so on. We provide a table of notation in Appendix A for review, and plan to expand it in the revision.   We will now address the further questions:
>
> 1 & 2. Thank you for catching this typo. $\gamma_h$ is indeed suppose to be mean zero and we will correct it in the revision.
> 3. When $\gamma_h$ is mean zero Gaussian, we recover the standard de-noising score matching loss where we add gaussian noise to a point and try to predict the noise. While this may seem unintuitive, many works, see for example [2] cited in our paper, show that this is equivalent to learning the score.
> 4. The distances $d_\mathcal{S}$ and $\mathsf{d}_{\mathrm{TVC}}$ are distances and thus real numbers, so the ordering is the standard one on the real line - x \vee y = max\{x,y\}. We will include this in the “notation” section.
>
> [1] C. Chi, S. Feng, Y. Du, Z. Xu, E. Cousineau, B. Burchﬁel, and S. Song. Diffusion policy:Visuomotor policy learning via action diffusion. arXiv preprint arXiv:2303.04137, 2023.
>
> [2]S. Chen, S. Chewi, J. Li, Y. Li, A. Salim, and A. Zhang. Sampling is as easy as learning the score: theory for
> diffusion models with minimal data assumptions. In NeurIPS 2022 Workshop on Score-Based Methods, 2022.

---

> > ### Comment · Reviewer_RM4v · 2023-08-17
> >
> > Thanks. I'm looking forward to the revised version.

---

### Official Review · Reviewer_NhZ6 · 2023-07-19

**Soundness:** 3 good
**Presentation:** 3 good
**Contribution:** 2 fair
**Rating:** 6
**Confidence:** 3

**Summary:**

This work proposes a theoretical framework for learning sequential decision-making with diffusion models. The authors show that with the conditions of (1) a suitable low-level "stability" guarantee and (2) a stochastic "continuity" property of the learned policy, the imitator can provably accurately estimate the "non-Markovian" policy of the demonstrators. The low-level "stability" is assumed in oracle primitive controllers, which are then composed by diffusion models that are learned with DDPM loss. Since the desired stochastic continuity is not guaranteed for an arbitrary demonstrator policy, the authors then propose a data augmentation technique that adds noise to the trajectories, making the resulting policies continuous. Their experiments validate that the proposed data augmentation alleviates imitator policy's degradation in noisy environments.

**Strengths:**

Originality: I am not very familiar with the current literature of learning theory. But proving the learnability of a hierarchical imitator appears to be novel to me.

Quality: The analysis presented in this paper is of high quality. I liked the formal narrative where each mathematical object is clearly defined before used. The authors also provide detailed derivation in their appendix.

Clarity: Even though this paper adopts heavy notation and formulas, the authors did a good job in explaining each mathematical object carefully. Through out my read, there are lines where I need to pause and think, but barely one makes me confused. The flow is very smooth.

Significance: I am not an expert in the literature of learning theory, so my evaluation of the significance may be biased. The proposed algorithm itself does not seem to be super significant. But conceptually connecting adding noises to "total variation continuity" is interesting to me.

**Weaknesses:**

My major concern is that even though the presented analysis is interesting and self-contained, it doesn't seem to provide too much insight into developing a new algorithm or improving existing ones. The proposed algorithm is a consequence of the analysis, but it is not obvious why we need it. I hope the authors would like to discuss more about the potential influence of the presented analysis and algorithm in their feedback.

**Questions:**

Apart from the question in Weakness, I hope the authors would like to answer the following:

L172, what is K? It does not seem to be defined.

I am not sure if this question is appropriate, but how can we know Assumptions 3.1, 3.2, and 3.3 are true in reality?

**Limitations:**

As stated by the authors in L239-243, the major limitation of this work is that the desired "stability" is given since the proposed algorithm relies on either synthesized primitive controllers (in our analysis) or low-level stabilizing controllers built into problem environments (in our experiments). Relaxing such restriction can be a future research direction.

---

> ### Author Rebuttal · Authors · 2023-08-09
>
> Thank you for your feedback.  Our approach suggests a natural algorithmic paradigm for imitation learning:
>
> Rather than imitating raw actions, one should imitate simple *control policies*. This paper addresses the simple case of linear gains, but in principle, one could imitate goals for goal-conditioned RL-trained reaching policies (we are working on this as an empirical follow-up). In the revision, we will better emphasize that this paper suggests a hierarchical algorithm design; we have been working on experience with synthesized gains for a 2D quadrotor to demonstrate this in the revised version.
> We show that adding smoothing noise at inference time if one adds noise at train time benefits performance.  While this intervention may seem small, it is important to note that our experiments show reasonable improvement over the baseline of adding the augmentation only at train time.
>
> In addition to potential algorithmic suggestions, our analysis explains the remarkable efficacy of diffusion policies imitating complex behavior in recent work [1,2,3]. As we stress in the introduction, no previous analysis can account for successful imitation of possibly multi-modal trajectories and our work is the first to do so.
>
> # Regarding The  Assumptions
> Asm 3.1 holds for many nonlinear systems - we are currently working with a collaborator to implement our hierarchical approach on a hydroelastic simulator of a robot with soft-end effectors, which satisfies this smoothness assumption (and experimentally has good sim-to-real transfer).
>
> As we remark after Asm 3.2, this holds whenever we synthesize gains from solving a Ricatti equation (and can be checked by inspecting the Ricatti value function).
> Asm 3.3 is a standard learning-theoretic assumption for “is learnable”, but as we show in a revision, a sufficient condition is simply that the DDPM has good validation loss (i.e. generalization under the expert distribution), which we show holds in practice.
>
> # Typos
>
>  1. The term $K_{k-1}$ in L172 should be $\bar K_{k}$.
>
> 2. On L169, we state that $$\bar{x}_k, \bar{u}_k,\bar{K}_k$$
> are feedback terms min a given sequence of primitive controllers.  We will correct this typo in the revision.
>
> #References
>
> [1] C. Chi, S. Feng, Y. Du, Z. Xu, E. Cousineau, B. Burchﬁel, and S. Song. Diffusion policy:Visuomotor policy learning via action diffusion. arXiv preprint arXiv:2303.04137, 2023.
>
> [2] Hansen-Estruch, I. Kostrikov, M. Janner, J. G. Kuba, and S. Levine. Idql: Implicit q-learning as an actor-critic method with diffusion policies. arXiv preprint arXiv:2304.10573, 2023
>
> [3] Pearce, T. Rashid, A. Kanervisto, D. Bignell, M. Sun, R. Georgescu, S. V. Macua, S. Z. Tan, I. Momennejad, K. Hofmann, et al. Imitating human behaviour with diffusion models. arXiv preprint arXiv:2301.10677, 2023

---

> > ### Comment · Reviewer_NhZ6 · 2023-08-17
> >
> > Thanks for your response. I am looking forward to the revision.

---

### Decision · Program_Chairs · 2023-09-21

**Decision:**

Accept (poster)

**Comment:**

The submitted paper was reviewed by 5 reviewers, 4 of whom recommended accpetance and 1 of whom recommended rejection. Discussions between reviewers and authors took place addressing most of the reviewers' questions. The reviewers appreciate the originality and rigour of the submitted paper but have concerns regarding the experimental evaluation, some of the made assumptions in the theoretical analysis, and the presentation ("complicated to parse", misconceptions/unclarities about the high-level implications and implications for future work). Based on my own reading of the paper, I also got the impression, that the paper could be significantly stronger and more impactful, if the authors simplify the presentation in the main paper and focus stronger on the high-level presentation, implications, and take-aways (for future work). Stil, I think the paper has sufficient merit to be accepted and hence I am recommending acceptance. Nevertheless, I encourage the authors to carefully revise their paper in line with the reviewers comments and issues highlighted in the dicussions, making the most out of the available 9 pages.